# Anchor-Based and Distributional Responsiveness of the Spanish Version of the Edinburgh Feeding Evaluation in Dementia Scale in Older People with Dementia: A Longitudinal Study

**DOI:** 10.3390/nu16223863

**Published:** 2024-11-12

**Authors:** Maria Carmen Saucedo-Figueredo, Juan Carlos Morilla-Herrera, María Gálvez-González, Francisco Rivas-Ruiz, Antonia Nava-DelVal, Mercedes San Alberto-Giraldos, Maria Jesús Hierrezuelo-Martín, Ana Belén Gómez-Borrego, Shakira Kaknani-Uttumchandani, José Miguel Morales-Asencio

**Affiliations:** 1IR Group C-13 “Chronicity, Dependency, Health Care and Services”, Malaga Biomedical Research Institute and Nanomedicine Platform (IBIMA BIONAND Platform), 29071 Malaga, Spain; shakira.kaknani@gmail.com (S.K.-U.); jmmasen@uma.es (J.M.M.-A.); 2Clinical Management Unit Los Boliches, Andalusian Health Service, District Costa del Sol, 29603 Malaga, Spain; jmorilla29@gmail.com; 3Nursing Home, Unit Málaga-Gualdalhorce, Andalusian Health Service, District Málaga, 29603 Malaga, Spain; 4Clinical Management Unit La Carihuela, Andalusian Health Service, District Costa del Sol, 29603 Malaga, Spain; maria.galvez.gonzalez.sspa@juntadeandalucia.es; 5Secretary Costa del Sol Research & Ethics Committee, Costa del Sol University Hospital, 29603 Malaga, Spain; francisco.rivas.ruiz.sspa@juntadeandalucia.es; 6Clinical Management Unit Las Albarizas, Andalusian Health Service District Costa del Sol, 29603 Malaga, Spain; manaval@hotmail.com; 7Clinical Management Unit La Lobilla, Andalusian Health Service District Costa del Sol, 29603 Malaga, Spain; mercedes.sanalberto.sspa@juntadeandalucia.es; 8Clinical Management Unit Las Lagunas, Andalusian Health Service District Costa del Sol, 29603 Malaga, Spain; mariaj.hierrezuelo.sspa@juntadeandalucia.es; 9Estepona Community, Mental Health Unit, Virgen de la Victoria University Hospital, 29010 Malaga, Spain; anab.gomez.borrego.sspa@juntadeandalucia.es; 10Faculty of Health Sciences, University of Málaga, 29071 Malaga, Spain

**Keywords:** dementia, feeding behaviour, longitudinal studies, predictive value of tests, malnutrition

## Abstract

Background/Objectives: Patients with dementia present with feeding difficulties (FDs) since diagnosis, conditioning their progression. Early identification is vital for preventing deterioration due to nutritional problems. The Edinburgh Feeding Evaluation in Dementia Scale (EdFED) identifies the FDs of patients with dementia by studying their behaviours while eating or being fed. The aim of this study was to assess the responsiveness of the EdFED over time in older people with cognitive impairment and to compare its effectiveness in identifying malnutrition and risk with that of the gold standard Mini Nutritional Assessment (MNA) method. Methods: This was a multicentre, prospective, observational, longitudinal, analytic study with a follow-up period of 18 months (with patients participating in nursing homes and in the community). Sociodemographic and nutritional data (body mass index (BMI), MNA, forearm circumference (FC), calf circumference (CC), and a nutritional blood test) were collected; EdFED score was reported by nurses, nursing assistants, and family caregivers. Results: The total sample consisted of 359 individuals—60.7% residential participants and 39.3% community participants. In the last follow-up there were 149 remaining (41.5%). Malnutrition was more than 30%, and the risk was 40% at the three follow-ups. The results suggest that the EdFED scale is a useful tool for assessing feeding difficulties (FDs) in older persons with dementia. It demonstrated good sensitivity and specificity in detecting malnutrition, similar to the MNA, and, more importantly, detecting risk and also identifying changes in nutritional status over time. Conclusions: The EdFED scale provides a means of evaluating nutritional problems, making it possible to work on prevention.

## 1. Introduction

The cognitive deficiencies and physical disabilities related to dementia often provoke feeding difficulties (FDs), which in turn have a negative impact on nutritional status and overall health [1,2].

Manifestations representative of FDs include the partial or complete inability to initiate or maintain attention to the action of eating, which includes putting food in the mouth, chewing or swallowing, or other eating behaviour problems, such as becoming distracted, slowing, refusing or showing apathy or indifference [3,4].

The above signs of FDs can be objectified, thus guiding clinicians towards the most appropriate form of intervention [1,5].

Some FDs are reversible, such as anorexia caused by infections or acute organ failure [6], while others can be managed by modifying environmental factors [7] or may improve if behavioural symptoms such as agitation, stress, or anxiety are controlled with medication [8].

FDs not only affect patients’ health but also have social consequences. Therefore, it is necessary to address and dignify the act of eating, an issue that is important both for institutions and for families [9].

FDs are most commonly identified through observation [10,11]. In this respect, the Edinburgh Feeding Evaluation in Dementia Scale (EdFED) (Appendix A) developed by Roger Watson offers good validity and reliability. This instrument has an intraclass correlation coefficient (ICC) of 0.44–1.00, a Cronbach’s alpha of 0.87, an intraobserver reliability of 0.95, and a validity of 0.85–0.90 [12,13]. The scale can be completed through direct observation or from the information gained in interviews with caregivers [14].

Among other versions that have been validated, the scale has been translated into Chinese [15,16,17], Italian [18,19], and Spanish [20,21]. The latter instrument has good internal consistency, with a Cronbach’s alpha of 0.88, a global interitem correlation of 0.43, and a homogeneity index of 0.42–0.73. Exploratory factor analysis of this scale reproduced the three-factor model identified by the original authors. Regarding criterion validity, the Spanish instrument reflects a moderately strong and statistically significant inverse correlation with serum albumin (ALB), total protein, and transferrin (TRF) [22,23]. Moreover, there was a good inverse correlation with the Mini Nutritional Assessment (MNA) [22,23] score and a moderate inverse correlation with body mass index (BMI). Higher EdFED values, which indicate a greater dependency on feeding, correlate with a greater risk of malnutrition and specifically with malnutrition identified by the MNA. The original EdFED scale is only applicable in nursing homes and must be implemented by nurses, but the Spanish-language version provides added value, as it can be administered by other health professionals (nursing assistants) and by family caregivers; furthermore, it can be used in other environments, such as daycare centres and in home care (healthcare in the community) [21].

The EdFED scale provides a means of evaluation that anticipates problems that may result in malnutrition and dysphagia and their consequences. The EdFED scale does not directly measure dysphagia but is designed to assess feeding problems in people with dementia. This scale identifies behaviours that may indicate difficulties with feeding, such as a loss of interest in food or the inability to coordinate feeding movements.

However, the condition of cognitive decline in these patients complicates the diagnosis of dysphagia, and they are often not referred to the hospital for specific tests, such as Fiberoptic Endoscopic Evaluation of Swallowing (FEES) or videofluoroscopic studies, which are considered the gold standard. From a behavioural perspective, these tests contribute little to the treatment plan. Instead, tools like the EAT-10 (Eating Assessment Tool [24]), which is used for dysphagia screening, are more useful in the context of dementia in geriatric patients. These tests provide sufficient data, and tests of different food textures help determine the best treatment for these patients [25]. However, the confirmatory diagnosis must be made as complete as possible, with techniques, tests, anthropometric examinations, and blood tests, within an interdisciplinary team and also with the help of algorithms [26].

The management of eating problems at an early stage, taking a new approach, could allow personalized care plans to be implemented both quickly and effectively, thus preventing or delaying FDs and their harmful consequences.

In this respect, Roger Watson carried out longitudinal studies in 1997 and 2017 [27,28] and observed a significant correlation between the EdFED score and the physical help given to patients. These studies confirmed the utility of the EdFED scale and identified its strengths and weaknesses, features that are probably replicated in other versions of the instrument. According to this researcher, the scale provides the most information at lower and higher levels of FD, with somewhat less information in the middle range. This finding indicated, for the first time, precisely where more work is needed on the scale, namely, the middle range. Thus, more items should be added to indicate or measure the presence of FDs. However, such items have yet to be determined in qualitative or observational research.

To our knowledge, no previous studies have evaluated the ability of the EdFED scale to identify longitudinal changes in nutritional status. Nevertheless, it is essential to understand this responsiveness to longitudinal change to confirm the evaluation strength of the scale and thus be able to compare it against the findings of other studies as a result variable for analytical and experimental designs. Such an evaluation would provide a more comprehensive overview of the patient’s nutritional status rather than a time-specific snapshot. Long-term follow-up would enable us to identify the extent to which FDs may be a predictor of malnutrition or undernutrition. With sufficient sensitivity to change over time, the scale could predict small changes in a patient’s nutritional status, even in the most advanced phases of dementia, with moderate–severe dependence despite the absence of major physical and/or psychological changes.

The study of responsiveness is a management tool that allows organisations to verify their achievements and to implement quality control. In short, the results obtained identify the interventions that are most effective for achieving the organization’s goals. In the healthcare environment, this means that better healthcare decisions can be made, resources can be allocated more appropriately, and attention can be provided more immediately.

In response to FDs, it is crucial to assess responsiveness in terms of the clinical use of the instrument to guide specific interventions aimed at improving behaviour and safety during feeding. Furthermore, when the EdFED scale result is used in research as an outcome variable for analytical and experimental designs, it is essential to understand the instrument’s ability to detect changes over time.

The aim of the present study was to assess the responsiveness of the Spanish version of the EdFED scale over time in an older population with moderate to severe cognitive impairment, both in residential care settings and in the community. The effectiveness of the MNA in identifying malnutrition and risk was compared with that of the gold standard method.

## 2. Materials & Method

This was a multicentre, prospective, observational, longitudinal, analytic study with a follow-up period of 18 months or prior to death.

### 2.1. Subjects

Resident participants of seven nursing homes and three Alzheimer’s daycare centres (ADCs) (Costa del Sol and Málaga Sanitary Districts) participated in this study, together with community-dwelling participants who lived exclusively in their own homes (7.8% of the study population). Follow-ups began in June 2021 and ended in December 2022.

The study population consisted of persons aged 65 years or older who had been institutionalized for at least three months, had been diagnosed with dementia (of any type), and who required assistance with feeding (Barthel index of activities for daily living, eating: ≤5). Individuals who were terminally ill or had additional illnesses that could hinder or prevent feeding (stroke, ALS, motor neuron diseases, Parkinson’s disease, fractures, paralysis, etc.); who had nasogastric, gastrostomy, or jejunostomy tubes; who required enteral nutrition; or whose legal guardians or responsible family members did not give consent for participation were excluded from the study.

The research was authorized to be carried out by the directorate of the Costa del Sol Health District on 27 February 2020. Funding was obtained on 15 February 2021 from the Junta de Andalucía by the Progress and Health Foundation (FPS) Project AP-0065-2020-C1-F2—‘Assessment of the Eating Pattern of People with Dementia in the Community through the Edinburgh Feeding Evaluation in Dementia Scale (EdFED). Longitudinal Follow-up study,’ corresponding to the announcement ‘FPS 2020—R&I projects in primary care, regional hospitals, and high-resolution hospitals (CHARES)’. It was also funded by the SEMILLA scholarship in 2023.

Approval by the Costa del Sol Biomedical Research Ethics Committee (CEI) came on 2 March 2021. Informed consent was obtained from all participants and collected until the beginning of the research in June 2021.

For each participant, the Spanish-language version of the EdFED scale was completed at recruitment and in three successive follow-ups, at average intervals of six months, during the 18-month follow-up. All follow-up evaluations were carried out by the same individuals, although this Spanish version of the instrument has previously demonstrated excellent interobserver reliability [21].

For the participants in the Alzheimer’s daycare centres and those in their own homes, three respondents (nurses, nursing assistants, and family caregivers) applied the EdFED scale based on their knowledge of the participating patients and the situation, thus providing three results for the same patient for three different meal settings—the breakfasts and lunches given at the centre and the dinners that, in every case, were consumed at home. In the case of the resident participants, only the nurse and the professional caregiver helped/supervised and were familiar with the participant’s eating habits and were able to respond to the questionnaire since the family caregivers were not identified, were not usually present, or did not usually participate in feeding the participants. Thus, two questionnaire responses were obtained for each participant regarding every meal consumed in the centre: breakfast, lunch, teatime snacks, and dinner.

Characteristics of the informants: nurses and nursing assistants—permanent staff with an average of 5 years of care in the same centre; informal caregivers—first-degree relatives (spouse, children) who live with the participant. They were the same people throughout all the follow-ups for each patient.

At each follow-up, a nutritional and anthropometric assessment was conducted to determine BMI, MNA, brachial circumference (FC), and calf circumference (CC). These data were collected at each follow-up. In addition, data on nutritional parameters (lymphocytes, absolute lymphocytes, total proteins, cholesterol, albumin, and transferrin) were collected at recruitment and at the last follow-up.

All data were collected or answered by three respondents (nurses, nursing assistants, and family caregivers) who applied the EdFED scale and completed the other questionnaires based on their knowledge of the participants and the situation, since the cognitive impairment that the patients presented prevented them from responding despite being able to speak in some cases.

The variables that were collected are described in the following table (Appendix A).

### 2.2. Sample Size

To detect an effect size of 0.2 in a repeated measures approach, with an alpha value of 0.05 and a power of 0.9, we calculated that 265 subjects would need to be recruited. This sample size was overestimated by 35% to allow for losses to study during follow-up. Furthermore, post hoc power analyses were conducted to test the impact of follow-up losses.

### 2.3. Analysis

An exploratory analysis was performed to obtain descriptive statistics of the variables. First, test–retest measures were calculated by the intraclass correlation coefficient (ICC) in a subsample of 97 participants (27%) to confirm the stability of the scores over a period of 48 h, the baseline period, during which no clinical changes occurred. Kappa interobserver analysis was carried out for individual items, concordance correlation coefficients (CCC) were determined, and the Bland–Altman test was conducted to obtain the total score.

Responsiveness was then evaluated by calculating distribution measures and by applying an anchor-based approach. In the first case, the minimum detectable change (MDC), effect size (ES), and standardized response mean (SRM) were estimated for the EdFED values obtained during the 18-month follow-up period. SRM values > 0.8 were considered indicative of high responsiveness, 0.5–0.8 reflected a moderate response, and <0.5 reflected a low response [29]. The MDC was evaluated as the smallest difference in EdFED scores, the ES was obtained using Cohen’s d, and the SRM was obtained by dividing the mean change in the score by the standard deviation of the change. Correlations between pairs of measures over time were estimated using the intraclass correlation coefficient (ICC).

In the anchor-based approach, the EdFED scores were compared among the participants whose MNA values indicated malnutrition, risk of malnutrition, or normal nutritional status. First, the linear correlation between the measures was calculated. Then, the differences in the mean values among the participants with normal nutritional status and those with risk of malnutrition were determined by ANOVA, with Brown–Forsythe homogeneity corrections. Games–Howell post hoc comparisons were performed to detect changes in the EdFED scores. Finally, the predictive value of the baseline EdFED score was calculated to compare the minimally important difference thresholds between participants who developed malnutrition at twelve months and those who did not, thus obtaining the corresponding ROC curves and determining the sensitivity and specificity of the scale [30]. All analyses were performed with the Jamovi 2.3 software program.

Since the missing data in the final follow-up were MCAR, complete case deletion was chosen for the analysis. For the T1 and T2 follow-ups, the impact of data loss was estimated using post hoc power calculations and discussed in the “Section 4”.

The baseline characteristics of the sample are described in Table 1.

## 3. Results

The total sample consisted of 359 individuals. In the final follow-up there were 149 remaining (41.5%). A total of 77 (21.4%) participants died before the first follow-up, 23 (6.4%) before the second follow-up, and 13 (3.6%) before the third follow-up. The total cumulative mortality of the sample was 31.5% (*n* = 113). Fourteen individuals (3.9% of the sample) were lost to the study because they were transferred to another centre, beyond the scope of this study. Losses in the third follow-up occurred due to circumstances unrelated to the participating subjects. Consequently, data could not be collected for 83 participants (23.1%) during this period. Accordingly, the third follow-up was not included in the responsiveness analysis. Analyses were performed until the second follow-up, which reached 247 patients. The cumulative drop-out rate was 58.5% (*n* = 210) at 18 months (Figure 1).

The age of the participants ranged from 65 to 105 years, with an average age of 82.9 years (SD 7.15). Of these participants, 86 (24%) were male and 273 (76%) were female. A total of 218 (60.7%) participants were permanently institutionalized, and 141 (39.3%) were enrolled in a daycare programme or lived at home (the community sample). Among the most prevalent pathologies, osteoarticular and vascular problems stand out. We must not lose sight of the figures for depression and falls. Twenty-eight percent were polypharmacy (five or more medications) (Table 1). Barthel showed severe physical dependence (35 points on average), and they presented moderate cognitive impairment approaching severe (mean of 7.6 points on the Pfeiffer). The severity of the dementia presented by the patients was high given the average values of the GDS (5.7). The nutritional parameters showed that they were mostly normal weight and with a BMI within normality (BMI 24%), the circumferences showed muscular mass within the limits: FC above 25 cm and CC above 31 cm. However, the mean MNA was 18.8 points, showing that the majority of the sample was at “risk of malnutrition”. It must be taken into account that 12% had dental problems, although only 2% had dysphagia and only 4% were benefiting from enteral feeding. Regarding the analytical parameters of malnutrition, an average of absolute lymphocyte of 1758 cel/mm^3^, cholesterol of 162.9 mg/dL, and albumin of 3.7 g/dL indicated mild malnutrition.

The profile of these participants, with the sample details, is summarized in Table 2.

The test–retest analysis yielded a concordance correlation coefficient of 0.85 (95% CI: 0.79–0.90). The Bland–Altman analysis (Figure 2) revealed a nonsignificant bias of 0.06 (95% CI: −0.30 to 0.42), and most of the differences in the scores versus the means were within the confidence intervals.

The FDs reflected by the EdFED scale scores remained largely unchanged throughout the follow-ups, as reported by the three types of responder (nurses, nursing assistants, and family caregivers). Table 3 details the variations in the EdFED scores for each follow-up period and by each type of evaluator who applied the scale.

These findings show that, for individual items, the Kappa agreement values ranged from 0.44 to 0.79, indicating moderate to substantial concordance among the evaluators. Regarding measures of change based on distributions, the most sensitive results for MDC and SRM were obtained in the assessments conducted between the first and second follow-ups (at 6 and 12 months, respectively) (Table 4).

Table 5 presents the outcomes for persons who were malnourished, at risk of malnutrition, and with a normal nutritional status during each of the follow-up periods, ranked according to the MNA stratification criteria, mid-upper arm circumference, leg circumference, BMI, and serum ALB. Due to losses during follow-up, the sample size evaluated differed among the participants. Malnutrition was more than 30%, and the average risk was over 40% at the three follow-ups.

The results of the sensitivity to change assessment based on the anchoring method in relation to the MNA showed that there were significant changes in EdFED scores between persons with a normal level of nutrition and those who presented alterations in their nutritional status. We obtained a mean difference of up to 8 points on the EdFED scale between these two groups, which represents 36% of the total range of the scoring system. Moreover, SMRs were over 0.5 and over 0.8 in most of the T2 follow-ups, indicating a high responsiveness in EdFED scores between patients without malnutrition and those with compromised nutritional status. These statements are based on SMR cut-off points reported in the literature [31,32]. When nutritional status was stratified according to the MNA criteria, the EdFED values assessed by nurses showed significant differences at baseline and at the 6- and 12-month follow-ups (up to 8 points). Similarly, large differences were observed in the assessments made by the nursing assistants and family caregivers. The highest SMR values were obtained between persons with a normal nutritional status and those with malnutrition, with values indicative of high responsiveness (>0.80). (Table 6).

Finally, the predictive validity of EdFED values for malnutrition or risk of malnutrition at baseline and at 12 months was assessed using the MNA. The ROC curve analysis (Figure 3) showed that there was an adequate predictive capacity for the baseline EdFED assessed by nurses for the prediction of risk of malnutrition or baseline malnutrition (area under the curve 0.81; *p* < 0.001), achieving, for a cut-off point of 6, a sensitivity of 67.55% and a specificity of 91.53%, with a positive predictive value (PPV) of 96.21% and a negative predictive value (NPV) of 46.96%. For the evaluations carried out by the nursing assistants, the PPV was 93.33% and the NPV was 42.86%, with an area under the curve of 0.82. For the presence or risk of malnutrition at 12 months with the MNA, with the same cut-off point of 6, the sensitivity was 63.64% and the specificity was 85.71%. For the assessments conducted by family caregivers, this predictive capacity decreased, with an area under the curve of 0.73. The optimal cut-off point was 5, with a sensitivity of 67.21%, a specificity of 72.22%, a PPV of 80.83%, and an NPV of 56.52%.

## 4. Discussion

In this study, the sample population was older and mostly female. The majority were living in nursing homes, with severe physical and cognitive impairment and provoked dependency. This level of dependency remained unaltered throughout the follow-up periods.

We consider it important to explain the losses in the sample before proceeding to discuss the results. Significant losses to follow-up occurred due to the number of dropouts and deaths among the sample population. However, according to the average Charlson Comorbidity Index for the participants in the sample, the expected mortality rate (MR) was 52% after three years, which is higher than the actual mortality rate. Alzheimer’s disease is the seventh leading cause of death worldwide, and its incidence increased by 55% between 1999 and 2014 [33,34]. In Spain, dementia causes 21,773 deaths a year. Alzheimer’s disease is the fifth leading cause of death, and mortality from this cause (affecting women in particular) has doubled in recent years [35,36].

The average life expectancy for people living with Alzheimer’s disease is four to eight years after diagnosis [37].

The main causes of mortality due to Alzheimer’s disease are concomitant infections (especially respiratory and urinary infections) caused by deterioration of the condition and overall patient fragility. Although these infections are considered the immediate cause of death, the underlying factor is dementia. The disease itself does not provoke death, but when, as a result, other systems fail, for example, the ability to swallow properly, this can lead to pneumonia when the patient aspirates secretions or food, with an outcome that is often fatal. From this clinical standpoint, FDs may increase the risk of pneumonia, hence worsening morbidity and mortality.

A certain degree of mortality is attributable to dementias per se. Although the presence of Alzheimer’s disease increases mortality, other forms of dementia can increase mortality [38,39,40].

With the present state of knowledge, early diagnosis does not change the course of the disease, but it does contribute to providing better attention, thus slowing the patient’s deterioration and preventing complications.

Despite the difficulties in distinguishing between different types of dementia, especially in advanced stages, an accurate diagnosis is crucial for understanding prognosis and guiding treatment [41,42]. Recent advances in biomarkers and imaging techniques have significantly improved diagnostic accuracy, which can lead to more specific and effective treatments [42].

However, some primary care physicians believe that refining the diagnosis through referrals to specialists for elderly patients with severe impairment does not provide significant benefits and can result in a lack of accurate diagnosis in the patient’s medical history. For this reason, we chose to include all subjects with dementia, as well as those in whom only cognitive decline was concluded.

The participants in our sample had been diagnosed with dementia for an average of 42 months (3.5 years) and presented high comorbidity and some FDs. Due to the increased risk of complications and mortality in this population, it would be useful in future studies to consider a larger sample, to recruit participants diagnosed at an earlier stage, and to increase the duration of follow-up (despite the evident difficulty posed in these respects).

In dementia, functional status depends on various factors in addition to cognition. Nutritional status is a potentially modifiable factor related to homeostasis and the proper functioning of body systems and can contribute to cognitive and functional impairment [43]. Therefore, it is important to have tools that enable the early detection and explanation of changes in nutritional status.

The EdFED scale has a maximum score of 12 points. In the present study, the sample produced a score slightly above the average, indicating that these participants mostly needed mild help and/or supervision to feed themselves or were moderately dependent in this respect.

The test–retest and Bland–Altman analyses indicated high consistency and reliability in the EdFED results, suggesting that this tool is suitable for measuring FDs in older persons with dementia. This conclusion is reinforced by the Kappa values obtained, which reflect moderate to substantial agreement among the evaluators regarding individual questionnaire items.

A study evaluating nutritional status and its association with behavioural psychiatric symptoms of dementia (BPSD) reported that nutritional status was significantly associated with specific BPSD, including “verbal aggression/emotional disinhibition” (F = 5.87, *p* = 0.016) and “apathy/memory impairment” (F = 15.38, *p* < 0.001), which were revealed by dementia behaviour disturbance (DBD) factor analysis [44]. The causal relationship between BPSD and nutritional problems is complex, but some BPSDs, such as apathy, can lead to loss of appetite and inactivity, resulting in weight loss and nutritional problems in patients with FDs. The severity of dementia when behavioural symptoms appear (GDS 6) is linked to nutritional status, being interrelated.

For a better understanding of this relationship, however, we need an instrument that is capable of evaluating behaviour patterns that directly alter nutritional status, thus obtaining timely relevant information from multiple sources. In this respect, we note that the EdFED scores remained stable over time and that the most significant changes occurred between follow-ups 1 and 2, which suggest that the questionnaire is sensitive to changes in FDs in this group of participants.

Sensitivity to change among longitudinal follow-up based on distributions were modest due to the stability of the scores. Moreover, the reduction in power due to losses in nursing assistants’ assessments suggests that findings in this subgroup should be taken with caution. The post hoc analysis revealed that follow-up losses affected the subgroups unevenly. While both the nurse and caregiver groups maintain adequate power, the nursing assistant group shows considerably low power, which limits the reliability of the results in this subgroup. This suggests that the findings related to nursing assistants should be interpreted with caution, as the remaining sample may not fully reflect the effect of the intervention. Furthermore, these results suggest that certain subgroups, such as nursing assistants, may be more vulnerable to losses or may exhibit variability in their interaction with patients. In light of this, we recommend that future studies consider increasing the initial sample size of these subgroups to ensure that, even with significant losses, adequate statistical power is maintained, thereby improving the reliability and generalizability of findings across each group of interest.

Conversely, in the anchor-based approach, the responsiveness of the EdFED score among groups according to nutritional status was very high. The SMR was greater than 0.80 in all groups of professionals and family caregivers, and the EdFED values were compared between subjects with a normal nutritional status and those with malnutrition, with higher values in comparisons in long-term follow-up. The instrument also presented good discriminative power between normal subjects and those at risk of malnutrition (SMR > 0.5 in all cases) and between at-risk subjects and those who were malnourished. From this finding, we conclude that when FDs are assessed with the EdFED scale, clinically significant, relevant differences can be observed among subjects according to their nutritional status. This observation is very important, both from a clinical perspective (the identification of FDs facilitates possible malnutrition) and for research purposes (EdFED is an outcome variable that is sensitive to changes in nutritional status). This conclusion was corroborated, with even greater robustness, by the high PPVs of baseline EdFED for malnutrition at 12 months: the PPV was >90% when used by nurses and was even >80% when carried out by family caregivers. EdFED, therefore, is readily applicable; it provides significant responsiveness to change for nutritional problems and can even be used by caregivers in various settings. Finally, the instrument can be used to obtain reliable long-term data.

For the relationship between the EdFED scores and nutritional status, the significant differences found between the participants with a normal nutritional status and those with nutritional alterations suggested that the questionnaire may be useful for identifying persons at risk of malnutrition.

The predictive capacity of the baseline EdFED score varied according to the evaluators involved; thus, the PPV was greater when the instrument was administered by nurses but decreased with the involvement of nursing assistants and family caregivers. This finding highlights the need to consider evaluators’ training and experience when using this questionnaire. Although the instrument has lower reliability when used by family caregivers, we consider it valuable to evaluate EdFED properties in this context since most patients in Spain are cared for by primary relatives, who have a thorough understanding of the patient status. If the EdFED scale is used by this group as a first-line tool for detecting problems, clinical nurses or other team members could then confirm the presence of feeding difficulties and initiate preventive measures.

This study has some limitations that should be acknowledged. First, when longitudinal measurements are obtained in such a sample of frail persons, a high rate of loss to follow-up is almost inevitable. The high mortality recorded during the short study period reflects the difficulty involved in obtaining follow-up data for such a vulnerable population. For this reason, the 12-month follow-up had to be omitted from the analysis, even with sample size overestimation.

Nevertheless, quality control standards were applied throughout this study to ensure good consistency in the measures obtained during the follow-up period. The data were always collected by the same persons, and the follow-up periods were respected, always taking into account the conditions applicable in each of the collaborating centres.

A further limitation is that anchor-based methods do not consider the measurement precision of the scale. Hence, a clinically significant change detected might fall within the range of random variation of the instrument. Moreover, the relationship between EdFED and MNA was not linear. To address this question, the linearity between the EdFED and MNA variables was previously verified. Finally, the study data were collected prospectively, thus avoiding biases derived from retrospective data collection.

In the MNA, answers for the questions, item O, ‘Self view of nutritional status?’ and item *p*, ‘In comparison with other people of the same age, how does the patient consider his/her health status?’ were answered by the professionals who collected the data or the family members (the same every time) who provided them given the patient’s level of cognitive impairment, comparing it with previous nutritional situations or with contemporaries.

## 5. Conclusions

In summary, these results suggest that the EdFED scale is a useful tool for assessing feeding difficulties in older persons with dementia by linking the behavioural symptoms present in dementia with nutritional problems and that it is especially effective when administered by trained healthcare professionals.

Compared with other tools for assessing nutritional standards, such as the MNA, the EdFED demonstrated good sensitivity and specificity in detecting malnutrition and, more importantly, detecting the risk of malnutrition, and it can also identify changes in nutritional status over time. It is essential to keep in mind that the final diagnosis of malnutrition or dysphagia should be made by interdisciplinary teams and based on comprehensive assessments, including various complementary tests.

More longitudinal and qualitative studies are needed to refine the capabilities of this tool, which is beginning to demonstrate important analytical qualities. Future lines of research could investigate which nursing interventions are most appropriate for each feeding difficulty encountered.

## Figures and Tables

**Figure 1 nutrients-16-03863-f001:**
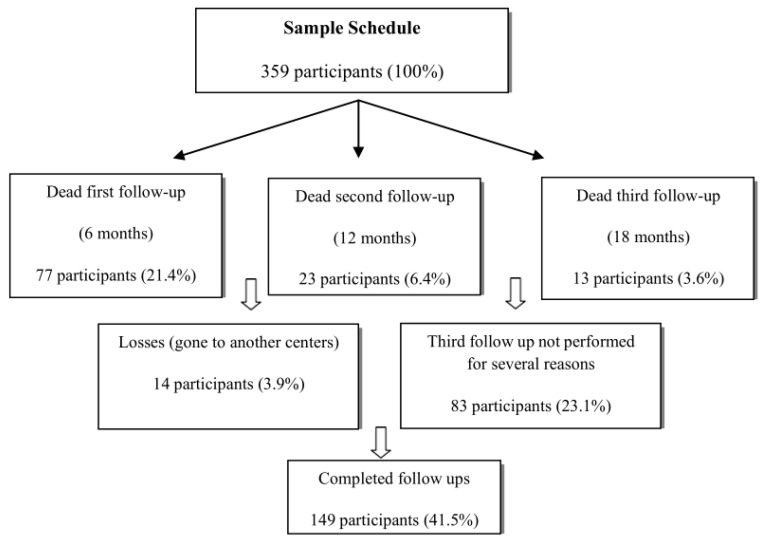
Sample schedule. Evolution of the sample of participants throughout the different follow-ups.

**Figure 2 nutrients-16-03863-f002:**
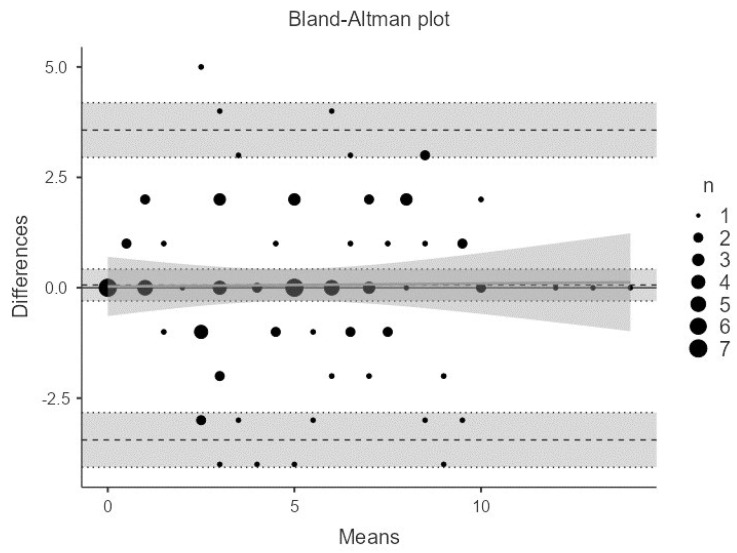
Bland–Altman plot analysis. Comparison of measurement techniques and evaluation of agreement.

**Figure 3 nutrients-16-03863-f003:**
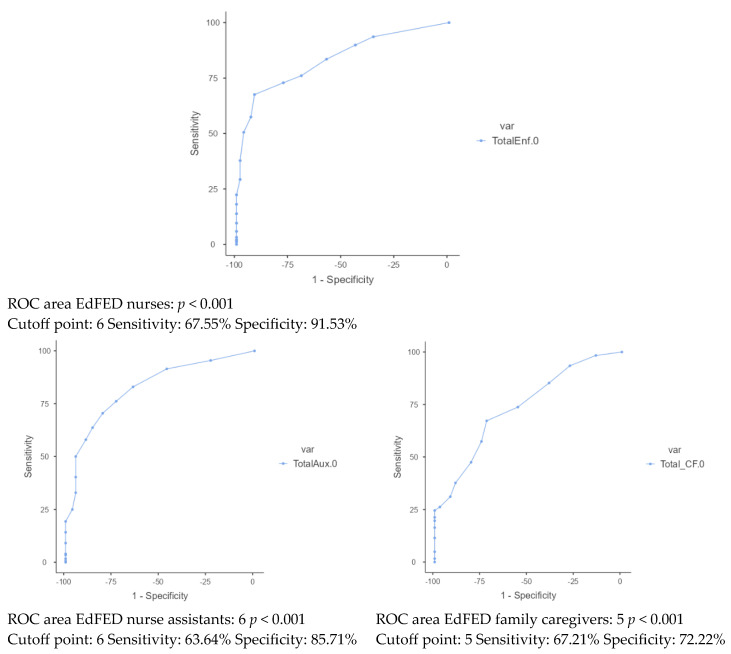
ROC curves. Baseline EdFED values assessed by nurses, nurse assistants, and family caregivers and risk of malnutrition at baseline or (evaluated with MNA) at 12 months.

**Table 1 nutrients-16-03863-t001:** Baseline characteristics. Descriptive and sociodemographic characteristics of the sample.

		Male (*n* = 86) *n* (%) or Mean (SD)	Female (*n* = 273) *n* (%) or Mean (SD)	Total *n* (%) or Mean (SD)	*p*
Setting					
	Nursing home	34 (39.5)	184 (67.4)	218 (60.7)	<0.001 *
	Home	13 (15.1)	15 (5.5)	28 (7.8)	0.016 *
	Day centre	39 (45.4)	74 (27.1)	113 (31.5)	0.475
Age (years)		81.4 (7.4)	83.5 (7.4)	83 (7.2)	
Dementia Type		Cognitive Impairment		216 (60.2%)	
		Alzheimer		113 (31.5%)	
		Vascular		9 (2.5%)	
		Mixed		19 (5.3%)	
		Fronto-Temporal		2 (0.5%)	
Time with dementia (months)		47.8 (39.7)	39.7 (40.8)	42.2 (32.5)	
(years)		4	3.3	3.5	
Health conditions					
	Osteoarticular	9 (10.5)	49 (17.9)	58 (16.2)	0.222
	Eye	9 (10.5)	31 (11.4)	40 (11.1)	0.234
	Ear	3 (3.5)	4 (1.5)	7 (1.9)	0.199
	Digestive	6 (7.0)	33 (12.1)	39 (10.9)	0.523
	Cancer	3 (3.5)	15 (5.5)	18 (5)	0.435
	Respiratory	8 (9.3)	12 (4.4)	20 (5.6)	0.026 *
	Diabetes	5 (5.8)	26 (9.5)	31 (8.6)	0.202
	Dyslipidemia	6 (7)	26 (9.5)	32 (8.9)	0.333
	Stroke	6 (7)	14 (5.1)	20 (5.6)	0.167
	Renal	6 (7)	13 (4.8)	19 (5.3)	0.463
	Depression	3 (3.5)	18 (6.6)	21 (5.8)	0.222
	Polymedication	24 (27.9)	78 (28.6)	102 (28.4)	0.514
	Falls	10 (11.6)	23 (8.4)	33 (9.2)	0.075

* scores that are statistically significant for that variable.

**Table 2 nutrients-16-03863-t002:** Nutritional, physical, and mental characteristics. Main results of the variables studied with respect to the level of physical and mental dependence and their nutritional status.

		Male (*n* = 86) *n* (%) or Mean (SD)	Female (*n* = 273) *n* (%) or Mean (SD)	Total *n* (%) or Mean (SD)	*p*
GDS *		5.54 (1.35)	5.72 (1.09)	5.68 (1.15)	0.497
Pfeiffer		7.2 (2.8)	8 (2.8)	7.8 (2.4)	0.009 *
Charlson		2.9 (1.6)	2.4 (1.6)	2.6 (1.6)	0.048 *
Barthel		40.6 (31.1)	31.1 (31.1)	33.4 (29.5)	0.009 *
MNA		19.6 (4.5)	18.3 (4.5)	18.6 (4.6)	0.017 *
Malnutrition		92 (33.6)	82 (33.2)	52 (34.9)	
Risk of malnutrition		129 (47.1)	106 (42.9)	65 (43.6)	
FC (cm)		25.7 (4.1)	25.1 (4.1)	25.2 (4.2)	0.300
CC (cm)		31.9 (4.2)	31 (4.2)	31.3 (4.6)	0.286
BMI (kg/m^2^)		24.4 (4.2)	23.8 (4.2)	24 (4.9)	0.272
Lymphocytes (%)		26.2 (8.7)	27.8 (8.7)	27.4 (9.3)	0.179
Lymphocytes (absolute) (Cel/mm^3^)		1727.6 (665.9)	1779.3 (665.9)	1766.9 (712.1)	0.560
Proteins (g/dL)		6.6 (0.7)	6.4 (0.7)	6.4 (0.6)	0.005 *
Cholesterol (mg/dL)		151.5 (33.9)	170.9 (33.9)	166.3 (35.1)	<0.001 *
Albumin (g/dL)		3.8 (0.4)	3.7 (0.4)	3.8 (0.4)	0.081
Transferrin (mg/dL)		222 (44.2)	232.7 (44.2)	230 (49)	0.104
Dental prostheses		13 (15.1)	27 (9.9)	40 (11.1)	0.060 *
Enteral nutrition		3 (3.5)	15 (5.5)	18 (5)	0.356
Dysphagia		2 (2.3)	6 (2.2)	8 (2.2)	0.591

* Baseline GDS (Global Deteriorating Scale), available in 104 subjects.

**Table 3 nutrients-16-03863-t003:** EdFED values for different evaluators. EdFED values evaluated by nurses, nursing assistants, and caregivers during follow-ups.

EdFED Nurses Mean (SD)		EdFED Nurse Assist Mean (SD)		EdFED Caregivers Mean (SD)	
Baseline (*n* = 359)	6.69 (4.63)	Baseline (*n* = 328)	6.90 (4.76)	Baseline (*n* = 139)	6.14 (4.34)
T1 (*n* = 275)	6.37 (5.05)	T1 (*n* = 258)	6.21 (4.87)	T1 (*n* = 108)	5.95 (4.33)
T2 (*n* = 248)	5.91 (4.68)	T2 (*n* = 233)	6.14 (4.88)	T2 (*n* = 94)	5.91 (4.46)

**Table 4 nutrients-16-03863-t004:** Measures of change based on distributions. Minimum detectable change (MDC), effect size (ES) of EdFED evaluated by different evaluators.

NURSES	Pre-Mean (SD)	Post-Mean (SD)	Mean Diff	Pooled SD	ICC	SEM	MDC	ES
T0-T1 (*n* = 247)	6.18 (4.56)	6.24 (5.02)	−0.1	1.53	0.8	0.11	**0.29**	**0.07**
T1-T2 (*n* = 247)	6.24 (5.02)	5.89 (4.68)	0.29	1.28	0.89	0.06	**0.16**	**0.23 ***
Systematic difference	F (1.8; 440) = 1.28; *p* = 0.277							
NURSING ASSISTANTS	Pre-Mean (SD)	Post-Mean (SD)	Mean diff	Pooled SD	ICC	SEM	**MDC**	**ES**
T0-T1 (*n* = 229)	6.21 (4.56)	6.10 (4.81)	0.11	1.08	0.74	0.14	**0.39**	**0.10**
T1-T2 (*n* = 229)	6.10 (4.81)	6.15 (4.88)	0.06	0.54	0.85	0.08	**0.22**	**0.11 ****
Systematic difference	F (1.8; 4.29) = 0.09; *p* = 0.905							
FAMILY CAREGIVERS	Pre-Mean (SD)	Post-Mean (SD)	Mean diff	Pooled SD	ICC	SEM	**MDC**	**ES**
T0-T1 (*n* = 92)	5.74 (4.48)	6.05 (4.51)	−0.31	0.37	0.84	0.08	**0.23**	**0.84**
T1-T2 (*n* = 92)	6.05 (4.51)	5.95 (4.50)	−0.11	0.21	0.81	0.10	**0.28**	**0.52 *****
Systematic difference	F (2.37; 118.8) = 3.44; *p* = 0.747							

Post hoc power: * Nurses, 0.95; ** Nursing assistants, 0.28; *** Family caregivers, 0.99.

**Table 5 nutrients-16-03863-t005:** Assessment of nutritional status over time. Distribution of nutritional status among the follow-up periods evaluated with different criteria: MNA, brachial circumference, leg circumference, BMI, and serum ALB.

MNA	Baseline * *n* (%) (*n* = 358)	6 Months *n* (%) (*n* = 274)	12 Months *n* (%) (*n* = 247)	*p*
Normal	52 (14.5)	53 (19.3)	59 (23.9)	0.05
Risk of malnutrition	169 (47.2)	129 (47.1)	106 (42.9)	
Malnutrition	137 (38.3)	92 (33.6)	82 (33.2)	
**Brachial circumference**	*n* (%) (*n* = 357)	*n* (%) (*n* = 276)	*n* (%) (*n* = 247)	
Normal	236 (66.1)	211 (76.4)	189 (76.5)	
Risk of malnutrition	88 (24.6)	38 (13.8)	34 (13.8)	0.002
Malnutrition	33 (9.2)	27 (9.8)	24 (9.7)	
**Leg circumference**	*n* (%) (*n* = 357)	*n* (%) (*n* = 275)	*n* (%) (*n* = 247)	
Normal	198 (55.5)	120 (43.6)	122 (49.4)	0.013
Malnutrition	159 (44.5)	155 (56.4)	125 (50.6)	
**BMI**	*n* (%) (*n* = 357)	*n* (%) (*n* = 275)	*n* (%) (*n* = 247)	
Normal/Overweight/Obesity	201 (56.3)	150 (54.5)	117 (47.4)	0.083
Risk of malnutrition	113 (36.3)	88 (36.1)	82 (37.1)	
Malnutrition	43 (13.8)	37 (15.2)	48 (21.7)	
**Serum albumin**	*n* (%) (*n* = 343)		*n* (%) (*n* = 252)	
Normal (3.5–5.3 mg/dL)	271 (79.0)		188 (74.6)	0.172
Slight malnutrition (2.8–3.4 mg/dL)	69 (20.1)		63 (25.0)	
Moderate malnutrition (2.8–3.4 mg/dL)	3 (0.9)		1 (0.39)	

* Baseline sample is not 359 patients (missing data due to impossibility of collection).

**Table 6 nutrients-16-03863-t006:** EdFED versus MNA. EdFED scores evaluated by nurses, nursing assistants, and family caregivers in subjects with normal versus altered nutritional status according to MNA cut-off points.

EdFED-NURSES		MD (95% CI; SMR)	MD (95% CI; SMR)
		MNA Risk of Malnutrition (*n* = 167)	MNA Malnutrition (*n* = 134)
Baseline (*n* = 353)	MNA Normal (*n* = 52)	−3.5 (−5.0 to −2.0; **0.88**)	−6.9 (−8.5 to −5.4; **1.74**)
	MNA Risk of malnutrition (*n* = 167)	—	−3.4 (−4.5 to −2.3; **0.85**)
		MNA Risk of malnutrition (*n* = 128)	MNA Malnutrition (*n* = 89)
6 months	MNA Normal (*n* = 53)	−3.3 (−4.5 to −2.1; **0.79**)	−7.7 (−9.2 to −6.3; **1.87**)
(*n* = 270)	MNA Risk of malnutrition (*n* = 128)	—	−4.5 (−6.0 to −3.0; **1.07**)
		MNA Risk of malnutrition (*n* = 105)	MNA Malnutrition (*n* = 80)
12 months (*n* = 244)	MNA Normal (*n* = 59)	−3.9 (−4.9 to −2.8; **1.08**)	−8.0 (−9.3 to −6.8; **2.24**)
	MNA Risk of malnutrition (*n* = 105)	—	−4.1 (−5.5 to −2.8; **1.16**)
**EdFED-NURSING ASSISTANTS**		MD (95% CI; **SMR**)	MD (95% CI; **SMR**)
		MNA Risk of malnutrition (*n* = 153)	MNA Malnutrition (*n* = 122)
Baseline	MNA Normal (*n* = 48)	−3.4 (−5.0 to −1.8; **0.82**)	−6.9 (−8.5 to −5.3; **1.67**)
(*n* = 323)	MNA Risk of malnutrition (*n* = 153)	—	−3.5 (−4.7 to −2.3; **0.85**)
		MNA Risk of malnutrition (*n* = 120)	MNA Malnutrition (*n* = 84)
6 months	MNA Normal (*n* = 49)	−2.5 (−4.1 to −0.8; **0.59**)	−6.6 (−8.3 to −5.0; **1.59**)
(*n* = 253)	MNA Risk of malnutrition (*n* = 120)	—	−4.2 (−5.7 to −2.6; **1.00**)
		MNA Risk of malnutrition (*n* = 97)	MNA Malnutrition (*n* = 76)
12 months	MNA Normal (*n* = 57)	−4.2 (−5.3 to −3.1; **1.13**)	−8.4 (−9.7 to −7.1; **2.29**)
(*n* = 230)	MNA Risk of malnutrition (*n* = 97)	—	−4.2 (−5.7 to −2.7; **1.15**)
**EdFED-FAMILY CAREGIVERS**		MD (95% CI; **SMR**)	MD (95% CI; **SMR**)
		MNA Risk of malnutrition (*n* = 64)	MNA Malnutrition (*n* = 33)
Baseline (*n* = 117)	MNA Normal (*n* = 40)	−2.2 (−3.9 to −0.5; **0.55**)	−4.9 (−6.9 to −2.8; **1.21**)
	MNA Risk of malnutrition (*n* = 64)	—	−2.7 (−4.7 to −0.6; **0.66**)
		MNA Risk of malnutrition (*n* = 53)	MNA Malnutrition (*n* = 17)
6 months (*n* = 108)	MNA Normal (*n* = 38)	−3.3 (−5.0 to −1.6; **0.92**)	−7.3 (−10.3 to −4.3; **2.03**)
	MNA Risk of malnutrition (*n* = 53)	—	−4.0 (−6.9 to −1.1; **1.11**)
		MNA Risk of malnutrition (*n* = 44)	MNA Malnutrition (*n* = 13)
12 months (*n* = 94)	MNA Normal (*n* = 37)	−3.2 (−5.2 to −1.2; **0.85**)	−7.5 (−10.4 to −4.7; **2.01**)
	MNA Risk of malnutrition (*n* = 44)	—	−4.3 (−7.2 to −1.5; **1.16**)

SMR: standardized response mean: MD, mean difference. The values in bold are the statistically significant ones. Throughout the text it is explained what values are considered within these for each measured parameter, in this case is SMR. In the text above “The highest SMR values were obtained between persons with a normal nutritional status and those with malnutrition, with values indicative of high responsiveness (>0.80)”.

## Data Availability

The research materials and datasets generated and/or analysed during the current study are available openly and free of charge in the repository of the library of the University of Malaga (RIUMA): https://riuma.uma.es/xmlui/ (accessed on 26 June 2024); https://dx.doi.org/10.24310/riuma.31733 (accessed on 26 June 2024).

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
