# Peer review of "Anchor-Based and Distributional Responsiveness of the Spanish Version of the Edinburgh Feeding Evaluation in Dementia Scale in Older People with Dementia: A Longitudinal Study"

_nutrients, 2024, doi:10.3390/nu16223863_

Round 1
Reviewer 1 Report
Comments and Suggestions for Authors
This paper aimed to assess the responsiveness of the EdFED in older people with cognitive impairment and to compare its effectiveness in identifying malnutrition risk with that of the gold standard MNA method.
Here my comments:
KEYWORDS: Please prefer MeSH keywords.
INTRODUCTION: Consider adding that FEES (Fiberoptic Endoscopic Evaluation of Swallowing) is often used as an adjunctive tool to better quantify dysphagia and its related risks.
METHODS: Would it be possible to include the causes of dementia in Table 1?
DISCUSSION: Please rewrite the beginning of the discussion, summarizing more clearly the key results of the study. Also, consider adding in the text the role of interdisciplinary approaches in diagnosing dysphagia across various neurological diseases (e.g., doi: 10.1016/j.otorri.2017.12.002).
Author Response
For research article
Anchor-based and distributional responsiveness of the Spanish version of the Edinburgh Feeding Evaluation in Dementia Scale in older people with dementia: a longitudinal study
Response to Reviewer 1 Comments
|
||
1. Summary |
|
|
Thank you very much for taking the time to review this manuscript. Please find the detailed responses below and the corresponding revisions/corrections highlighted/in track changes in the re-submitted files.
|
2. Questions for General Evaluation |
Reviewer’s Evaluation |
Response and Revisions |
Does the introduction provide sufficient background and include all relevant references? |
Yes/Can be improved/Must be improved/Not applicable |
[Please give your response if necessary. Or you can also give your corresponding response in the point-by-point response letter. The same as below] |
Are all the cited references relevant to the research? |
Yes/Can be improved/Must be improved/Not applicable |
|
Is the research design appropriate? |
Yes/Can be improved/Must be improved/Not applicable |
|
Are the methods adequately described? |
Yes/Can be improved/Must be improved/Not applicable |
|
Are the results clearly presented? |
Yes/Can be improved/Must be improved/Not applicable |
|
Are the conclusions supported by the results? |
Yes/Can be improved/Must be improved/Not applicable
|
|
3. Point-by-point response to Comments and Suggestions for Authors
|
||
Comments 1: [Paste the full reviewer comment here.]
|
This paper aimed to assess the responsiveness of the EdFED in older people with cognitive impairment and to compare its effectiveness in identifying malnutrition risk with that of the gold standard MNA method.
Here my comments:
KEYWORDS: Please prefer MeSH keywords.
INTRODUCTION: Consider adding that FEES (Fiberoptic Endoscopic Evaluation of Swallowing) is often used as an adjunctive tool to better quantify dysphagia and its related risks.
METHODS: Would it be possible to include the causes of dementia in Table 1?
DISCUSSION: Please rewrite the beginning of the discussion, summarizing more clearly the key results of the study. Also, consider adding in the text the role of interdisciplinary approaches in diagnosing dysphagia across various neurological diseases (e.g., doi: 10.1016/j.otorri.2017.12.002).
Response 1
Response to the reviewer’s comment on KEYWORDS
Thank you for your observation regarding the keywords. I would like to confirm that all the terms used in the manuscript are codified in the MeSH (Medical Subject Headings) system, as used in PubMed. Below, I provide the terms and their corresponding Unique Identifiers (UIDs) for your verification:
Dementia – MeSH Unique ID: D003704
Feeding Behavior – MeSH Unique ID: D005270
Longitudinal Studies – MeSH Unique ID: D017418
Predictive Value of Tests – MeSH Unique ID: D011241
Malnutrition – MeSH Unique ID: D008393
These terms have been selected in accordance with MeSH guidelines to ensure proper indexing and search ability in scientific databases.
INTRODUCTION
We did not consider using the FEES (Flexible Endoscopic Evaluation of Swallowing) procedure to diagnose dysphagia in these patients because, given their condition, it is difficult for them to attend a hospital to undergo this test. FEES cannot be performed at home, in care homes, or in Alzheimer's day centres, since it is a hospital-based technique. Additionally, our sample did not attend the nutrition unit in the hospital. Including a reference to FEES might confuse the reader, as the objective of our study was not to confirm dysphagia in these patients through exploratory tests. Furthermore, addressing this would require discussing the broader methods of diagnosing malnutrition and dysphagia, along with various confirmatory tests, rather than focusing solely on FEES. As an alternative, we aimed to concentrate on the EdFED scale, which is the primary objective of our research."Amendment in the text; (Page 2, last paragraph, line 48):“The EdFED scale does not directly measure dysphagia, but is designed to assess feeding problems in people with dementia. This scale identifies behaviors that may indicate difficulties with feeding, such as a loss of interest in food or the inability to coordinate feeding movements.However, the condition of cognitive decline in these patients complicates the diagnosis of dysphagia, and they are often not referred to the hospital for specific tests, such as Fiberoptic Endoscopic Evaluation of Swallowing (FEES) or videofluoroscopic studies, which are considered the gold standard. From a behavioral perspective, these tests contribute little to the treatment plan. Instead, tools like the EAT-10 (Eating Assessment Tool), which is used for dysphagia screening, are more useful in the context of dementia in geriatric patients. These tests provide sufficient data, and tests of different food textures help determine the best treatment for these patients (24)”.https://doi.org/10.1016/j.clnu.2024.04.039
METHODS
We have included this information in table 1: We chose as an inclusion criterion to have the diagnosis of Dementia recorded in their health history in any of its forms, including cognitive impairment. However, there are many patients who are diagnosed with cognitive impairment or senile dementia by their primary care physicians because at those ages it is not confirmed by the neurologist. I add in table 1;
Dementia Type |
Cognitive Impairment |
216 (60.2%) |
Alzheimer |
113 (31.5%) |
|
Vascular |
9 (2.5%) |
|
Mixed |
19 (5.3%) |
|
Fronto-Temporal |
2 (0.5%) |
“Despite the difficulties in distinguishing between different types of dementia, especially in advanced stages, an accurate diagnosis is crucial for understanding prognosis and guiding treatment (37,38). Recent advances in biomarkers and imaging techniques have significantly improved diagnostic accuracy, which can lead to more specific and effective treatments (38).
However, some primary care physicians believe that refining the diagnosis through referrals to specialists for elderly patients with severe impairment does not provide significant benefits, which can result in a lack of accurate diagnosis in the patient's medical history. For this reason, we chose to include all subjects with dementia, as well as those in whom only cognitive decline was concluded”.
DISCUSSIONWe have structured the discussion with an initial section focused on explaining the findings related to the characteristics of our study sample and comparing these with current epidemiologic data on the population with dementia, as well as the implications for patient’s progression. This aspect is essential since our study had a longitudinal design, and the advanced stage of functional impairment, together with a high prevalence of comorbidities must be considered when interpreting results. Following, we highlight the importance of having tools to detect the early onset of changes in nutritional status, in this case through patients behaviours, assessed using the EdFED scale. Consequently, we discuss the findings on reliability, which showed a robust data regarding inter-rater agreement and consistency of scores in a short period of time. In the next section of the discussion we discuss the implications of our findings related to sensitivity to change, based on SMR effect sizes. Moreover, we have included a cautionary comment regarding the limited power for the nursing assistants subgroup, based on post-hoc power analyses.
The discussion also follows the same sequence as the results to explain them in the same order, except for that explanation at the beginning of the unexpected losses in the sample that we consider essential to introduce before proceeding to discuss the results in depth.
Nevertheless, we have further explained the findings for each objective by adding new paragraphsAnd with respect to the point we introduced at the beginning "losses", the following amendment has been done to improve understanding:(Page 14, paragraph 3, line 7) “We consider it important to explain the losses in the sample before proceeding to discuss the results“
On the other hand, we have considered the role of interdisciplinarity, which is reflected in the research with a separate analysis of the different respondents (nurse, nursing assistants and family caregivers (table 3). The suggested reference focuses on dysphagia and neurodegenerative conditions, while our study on the EdFED scale examines altered behaviours that nutritionally impact the patient, which may but do not exclusively, lead to dysphagia. We thought that including this reference could divert the reader’s attention from the intended focus of our results.
Nevertheless, the following text has been included in the article:(Page 17, paragraph 3, line 15 “Conclusions”)
“It is essential to keep in mind that the final diagnosis of malnutrition or dysphagia should be made by interdisciplinary teams and based on comprehensive assessments, including various complementary tests.”
This addition also responds to your request at INTRODUCTION that other diagnostic imaging tests such as FEED should confirm the diagnosis.

Reviewer 2 Report
Comments and Suggestions for Authors
The manuscript studied the validity of applying the EdFED scale to evaluate FDs in dementia patients. However, there are several concerns about this paper.
1. Dropout was quit high (58.5%). The authors should elaborate on strategies for addressing this limitation or employ more robust statistical adjustments.
2. The methodology lacks clarity on procedures like missing data handling.
3. The discussion section overinterprets the findings, drawing broad conclusions from a limited dataset.
4. The authors claim that the EdFED scale is effective at detecting both malnutrition and its risks. However, the MDC and SRM presented moderate to low responsiveness for many of the evaluators, particularly between baseline and first follow-up.
5. For family caregivers, their clinical accuracy may be less reliable than that of professionals.
Author Response
For research article
Anchor-based and distributional responsiveness of the Spanish version of the Edinburgh Feeding Evaluation in Dementia Scale in older people with dementia: a longitudinal study
Response to Reviewer 2 Comments
|
||
1. Summary |
|
|
Thank you very much for taking the time to review this manuscript. Please find the detailed responses below and the corresponding revisions/corrections highlighted/in track changes in the re-submitted files.
|
2. Questions for General Evaluation |
Reviewer’s Evaluation |
Response and Revisions |
Does the introduction provide sufficient background and include all relevant references? |
Yes/Can be improved/Must be improved/Not applicable |
[Please give your response if necessary. Or you can also give your corresponding response in the point-by-point response letter. The same as below] |
Are all the cited references relevant to the research? |
Yes/Can be improved/Must be improved/Not applicable |
|
Is the research design appropriate? |
Yes/Can be improved/Must be improved/Not applicable |
|
Are the methods adequately described? |
Yes/Can be improved/Must be improved/Not applicable |
|
Are the results clearly presented? |
Yes/Can be improved/Must be improved/Not applicable |
|
Are the conclusions supported by the results? |
Yes/Can be improved/Must be improved/Not applicable
|
|
3. Point-by-point response to Comments and Suggestions for Authors
|
||
Comments 1: [Paste the full reviewer comment here.]
|
The manuscript studied the validity of applying the EdFED scale to evaluate FDs in dementia patients. However, there are several concerns about this paper.
- Dropout was quit high (58.5%). The authors should elaborate on strategies for addressing this limitation or employ more robust statistical adjustments.
- The methodology lacks clarity on procedures like missing data handling.
- The discussion section over interprets the findings, drawing broad conclusions from a limited dataset.
- The authors claim that the EdFED scale is effective at detecting both malnutrition and its risks. However, the MDC and SRM presented moderate to low responsiveness for many of the evaluators, particularly between baseline and first follow-up.
- For family caregivers, their clinical accuracy may be less reliable than that of professional
Response 1
1. The sample size calculation was based on prevalence data available in the literature, indicating that a sample of 265 patients was needed to achieve adequate statistical power for follow-up. We even increased the sample by 35% to account for anticipated losses. However, the losses exceeded our projections, highlighting the extreme fragility of the dementia population and the limitations of the initial estimates, especially concerning morbidity and mortality. We have emphasised this issue in the discussion section, providing a detailed explanation to guide future researchers studying dementia populations in adjusting their sample loss estimates, potentially up to twice the percentage suggested in the literature at the time.(Page 5, last paragraph, last line) “Losses in the third follow-up occurred due to circumstances unrelated to the participating subjects. Consequently, data could not be collected for 83 participants (23.1%) during this period. Accordingly, the third follow-up was not included in the responsiveness analysis. The analyzers are done till the second follow-up, which reached 247 patients.”
(Page 6, first paragraph, line 3)
“Analyzes were done till the second follow-up, which reached 247 patients.”
However, analyzes were done until the second follow-up (12 months) to which 247 patients arrived. The losses were not so far from the initial sample (18 patients), thus preserving their statistical potential.
We have also included post-hoc power calculations to estimate the impact of this shortfall. For the obtained effect sizes in each subgroup of evaluators, the power calculations yielded:Nurses: ES 0.23; post-hoc power 0.95Nursing assistants: ES 0.11; post-hoc power 0.28Caregivers: ES 0.52 post-hoc power 0.99We have included this information both in the analysis section and as a footnote in Table 3. (Page 5, second paragraph, line 9)“Furthermore, post-hoc power analyses were conducted to test the impact of follow-up losses.” (Page 5, paragraph 6, line 41)“Since the missing data in the final follow-up were MCAR, complete case deletion was chosen for the analysis. For the T1 and T2 follow-ups, the impact of data loss was estimated using post-hoc power calculations and discussed in the discussion section.” Moreover, we have included cautionary comments of the impact of this issue in the nursing assistants subgroup. We add in DISCUSSION (Page 15, paragraph 9, line 46)“The post hoc analysis reveals that follow-up losses affected the subgroups unevenly. While the nurses and caregivers groups maintain adequate power, the nursing assistants group shows considerably low power, which limits the reliability of the results in this subgroup. This suggests that the findings related to nursing assistants should be interpreted with caution, as the remaining sample may not fully reflect the effect of the intervention. Furthermore, these results suggest that certain subgroups, such as nursing assistants, may be more vulnerable to losses or may exhibit variability in their interaction with patients. In light of this, we recommend that future studies consider increasing the initial sample size of these subgroups to ensure that, even with significant losses, adequate statistical power is maintained, thereby improving the reliability and generalizability of findings across each group of interest. 2. A paragraph has been included in the analysis section to describe how missing data were managed. (Page 5, paragraph 6, line 41)
“Since the missing data in the final follow-up were MCAR, complete case deletion was chosen for the analysis. For the T1 and T2 follow-ups, the impact of data loss was estimated using post-hoc power calculations and discussed in the discussion section.”
(See also the explanbation in previous paragraph 1) 3 and 4. We have structured the discussion with an initial section focused on explaining the findings related to the characteristics of our study sample and comparing these with current epidemiologic data on the population with dementia, as well as the implications for patient’s progression. This aspect is essential since our study had a longitudinal design, and the advanced stage of functional impairment, together with a high prevalence of comorbidities must be considered when interpreting results. Following, we highlight the importance of having tools to detect the early onset of changes in nutritional status, in this case through patients behaviours, assessed using the EdFED scale. Consequently, we discuss the findings on reliability, which showed a robust data regarding inter-rater agreement and consistency of scores in a short period of time. In the next section of the discussion we discuss the implications of our findings related to sensitivity to change, based on SMR effect sizes. Moreover, we have included a cautionary comment regarding the limited power for the nursing assistants subgroup, based on post-hoc power analyses. With regard to sensitivity to change based on distributional methods, effectively, the results were discrete. We have introduced a paragraph in the discussion to put emphasis on this issue, even warning on the power limitation in the nursing assistants group. (Page 15, paragraph 10, line 46)
“Sensitivity to change among longitudinal follow-up based on distributions were modest due to the stability of the scores. Moreover, the reduction in power due to losses in nursing assistants’ assessments suggests that findings in this subgroup should be taken with caution.”
On the other hand, the anchor methods showed discriminatory capacity of EdFED between subjects with normal nutritional status and those whose nutritional status is affected. We obtained a mean difference up to 8 points on the EdFED scale between these two groups, which represents a 36% of the total range of the scoring system. Moreover, SMR were over 0.5 and over 0.8 in most of T2 follow-ups, indicating a high responsiveness in EdFED scores between patients without malnutrition and those with compromised nutritional status. These statements are based on SMR cut-off points reported in the literature (Morrow et al., 2016; Cornett et al. 2020). We do not consider this inference to be overestimated, as we use accepted reference criteria to estimating our findings. (Page 11, second paragraph, line 7) Finally, we highlight some cautions regarding our findings and limitations associated with our study. “Conversely, in the anchor-based approach, the responsiveness of the EdFED score among groups according to nutritional status was very high. The SMR was greater than 0.80 in all groups of professionals and family caregivers and the EdFED values were compared between subjects with a normal nutritional status and those with malnutrition, with higher values in comparisons in long-term follow-up.”(Page 16, paragraph 2, line 11) 5. We agree with the fact that the instrument has lower reliability when used by family caregivers, however, we consider it valuable to evaluate EdFED properties in this context since most patients in Spain are cared for by primary relatives, who have a thorough understanding of the patient status. If the EdFED scale is used by this group as a first-line tool for detecting problems, clinical nurses or other team members could then confirm the presence of feeding difficulties and initiate preventive measures. We have added this text in the discussion.(Page 16, paragraph 4, line 32) .

Round 2
Reviewer 1 Report
Comments and Suggestions for Authors
Here my comments:
-
Clarification on FEES: The Fiberoptic Endoscopic Evaluation of Swallowing (FEES) is not exclusively a hospital-based procedure; it can also be included in bedside examinations. Clarifying this point will provide readers with a comprehensive overview of techniques for assessing swallowing disorders, and eventual consequent malnutrition, ensuring they do not mistakenly believe FEES is confined to hospital settings. This also sets the stage for the study's focus on the Edinburgh Dysphagia Score.
-
Suggestion on Scale Inclusion: Since the focus is on the Edinburgh Dysphagia Score, I suggest introducing an additional table listing the most commonly used validated scales that can be used in dysphagia assessment for dementia patients. This addition would provide valuable context, particularly for less experienced readers, and enhance understanding of the scales within the broader scope of dysphagia/malnutrition assessment in dementia.
-
On EdFED Scale Purpose: You noted that "The EdFED scale does not directly measure dysphagia but is designed to assess feeding problems," that can cause malnutrition, which aligns well with the study's focus. To enhance the introduction, consider concluding with a brief mention of diagnostic algorithms always more proposed for nutritional disorders, which can predict malnutrition and even muscle mass loss (DOI: 10.1016/j.heliyon.2023.e16323).
- Interdisciplinary Approach in Conclusion: In the conclusion, the authors highlight the importance of an interdisciplinary approach in managing swallowing disorders. However, this recommendation lacks sufficient support within the main text. It would strengthen the paper to include a discussion or references earlier in the text, as suggested in my previous revision, that justify this interdisciplinary need by illustrating the contributions of various professionals in comprehensive dysphagia assessment and management also in different disorders.
Author Response
For research article
Anchor-based and distributional responsiveness of the Spanish version of the Edinburgh Feeding Evaluation in Dementia Scale in older people with dementia: a longitudinal study
Response to Reviewer 1 Comments 2
|
||
1. Summary |
|
|
Thank you very much for taking the time to review this manuscript. Please find the detailed responses below and the corresponding revisions/corrections highlighted/in track changes in the re-submitted files.
|
2. Questions for General Evaluation |
Reviewer’s Evaluation |
Response and Revisions |
Does the introduction provide sufficient background and include all relevant references? |
Yes/Can be improved/Must be improved/Not applicable |
[Please give your response if necessary. Or you can also give your corresponding response in the point-by-point response letter. The same as below] |
Are all the cited references relevant to the research? |
Yes/Can be improved/Must be improved/Not applicable |
|
Is the research design appropriate? |
Yes/Can be improved/Must be improved/Not applicable |
|
Are the methods adequately described? |
Yes/Can be improved/Must be improved/Not applicable |
|
Are the results clearly presented? |
Yes/Can be improved/Must be improved/Not applicable |
|
Are the conclusions supported by the results? |
Yes/Can be improved/Must be improved/Not applicable
|
|
3. Point-by-point response to Comments and Suggestions for Authors
|
||
Comments 2: [Paste the full reviewer comment here.]
|
- Clarification on FEES: The Fiberoptic Endoscopic Evaluation of Swallowing (FEES) is not exclusively a hospital-based procedure; it can also be included in bedside examinations. Clarifying this point will provide readers with a comprehensive overview of techniques for assessing swallowing disorders, and eventual consequent malnutrition, ensuring they do not mistakenly believe FEES is confined to hospital settings. This also sets the stage for the study's focus on the Edinburgh Dysphagia Score.
- Suggestion on Scale Inclusion: Since the focus is on the Edinburgh Dysphagia Score, I suggest introducing an additional table listing the most commonly used validated scales that can be used in dysphagia assessment for dementia patients. This addition would provide valuable context, particularly for less experienced readers, and enhance understanding of the scales within the broader scope of dysphagia/malnutrition assessment in dementia.
- On EdFED Scale Purpose: You noted that "The EdFED scale does not directly measure dysphagia but is designed to assess feeding problems," that can cause malnutrition, which aligns well with the study's focus. To enhance the introduction, consider concluding with a brief mention of diagnostic algorithms always more proposed for nutritional disorders, which can predict malnutrition and even muscle mass loss (DOI: 10.1016/j.heliyon.2023.e16323).
- Interdisciplinary Approach in Conclusion: In the conclusion, the authors highlight the importance of an interdisciplinary approach in managing swallowing disorders. However, this recommendation lacks sufficient support within the main text. It would strengthen the paper to include a discussion or references earlier in the text, as suggested in my previous revision, that justify this interdisciplinary need by illustrating the contributions of various professionals in comprehensive dysphagia assessment and management also in different disorders.
Response
- Although it is a technique that can be performed at the patient's bedside at home. In Spain, this technique is not in the portfolio of primary care services. Therefore this possibility does not exist in our country. As we have already explained in our previous comment answers.
Doctors do not consider it ethical to send these patients to the hospital given their condition to perform confirmatory tests for dysphagia when the care plan is going to be the same (thickeners, diet adaptation, etc.) more aimed at reducing even more risk. of health.
It is explained in the text (Page 3, paragraph 1, line 1);
“However, the condition of cognitive decline in these patients complicates the diagnosis of dysphagia, and they are often not referred to the hospital for specific tests, such as Fiberoptic Endoscopic Evaluation of Swallowing (FEES) or videofluoroscopic studies, which are considered the gold standard.”
2. There are no specific tools to diagnose dysphagia in adults with dementia, several are used and they continue to investigate which one has the greatest validity and reliability. In a current study (doi: 10.1007/s00455-024-10707-0) they used these three, being the best the first;a. The Oropharyngeal Dysphagia Screening Test for Patients and Professionals (ODS-PP) [29)b. The Eating Assessment Tool-10 (EAT-10) [36, 37]c. The Swallowing Disturbance Questionnaire (SWAL-QOL) [38,39,40,41] Another attempt to validate scales for dysphagia in dementia is the Caregiver Questionnaire – RaDID-QC. doi: 10.1016/j.clinsp.2024.100440 They are always scales and not invasive procedures through information from caregivers as we did in our article doi.org/10.1371/journal.pone.0192690 The same occurs with Desnutrition. The MNA is the only than has been validated but saying for any adults including these with dementia (Kaiser M.J.et al. 2009).
The diagnosis confirmation for dysphagia may include the following:
- Administration of an interview or a questionnaire that addresses the patient’s perception of and/or concern with swallowing function (e.g., the 10-item Eating Assessment Tool [EAT-10]; Cheney, 2015).
- Monitoring the presence of the signs and symptoms of oropharyngeal and/or esophageal swallowing dysfunction.
- Patient/caregiver report or observation of difficulty with per os (P.O.) intake.
- Administration of standardized screening protocols, such as
- the 3-oz water swallow test (DePippo et al., 1992) and
- the Yale Swallow Protocol (Suiter et al., 2014).
- Administration of the modified Evans blue dye test in patients with a tracheotomy by tinting oral feedings blue/green with the intent to identify aspiration in these patients (Béchet et al., 2016). For further information on the modified Evans blue dye test, please see the FDA public health advisory.
Other authors focus on knowing which test is best for diagnosing dysphagia but in neuromuscular diseases (NMD). The latest systematic reviews show that dysphagia in patients with neuromuscular problems is better identified with non-invasive tests.
Table.Tools used to study dysphagia in each NMD
Underlying diseases |
Tools |
Study |
ALS |
VFSS, sEMG (DL), FEES, V-VST, Man., VCA, 3SwT, NdSSS, MTP, EAT-10 |
Murono and colleagues55; Aydogdu and colleagues14; Mari and colleagues47; Paris and colleagues46; Plowman and colleagues49,50,62; Kidney and colleagues48; Wada and colleagues41; Briani and colleagues10; Cosentino and colleagues54; Hiraoka and colleagues53; Plowman and colleagues49,50,62 |
DMD |
VFSS, SSQ, sEMG, NdSSS |
Archer and colleagues43; Archer and colleagues44; Hanayama and colleagues42; Wada and colleagues41 |
DM1 |
FEES, sEMG (DL), 3SwT |
Pilz and colleagues45; Aydogdu and colleagues14; Mari and colleagues47 |
MG |
sEMG (DL), VFSS |
Higo and colleagues56; Aydogdu and colleagues14 |
IBM |
Standard Questionnaire, RT-MRI |
Cox and colleagues7; Olthoff and colleagues51 |
SMA |
VFSS, Man., FEES |
Briani and colleagues10 |
PM/DM |
sEMG (DL) |
Aydogdu and colleagues14 |
FA |
3SwT |
Mari and colleagues47 |
SBMA |
MTP |
Mano and colleagues52 |
3SwT, 3-ounce water swallow test; ALS, amyotrophic lateral sclerosis; DL, dysphagia limit; DM1, myotonic dystrophy type 1; DMD, Duchene muscular dystrophy; EAT-10, eating assessment tool; FA, Friedreich’s ataxia; FEES, fiberoptic endoscopic evaluation of swallowing; IBM, inclusion body myositis; Man., pharyngo-esophageal manometry; MG, myasthenia gravis; MTP, maximum tongue pressure; NdSSS, neuromuscular disease swallowing status scale; NMD, neuromuscular disease; PM/DM, polymyositis/dermatomyositis; RT-MRI, real-time magnetic resonance imaging; SBMA, spinal and bulbar muscular atrophy; sEMG, surface electromyography; SMA, spinal muscular atrophy; SSQ, Sydney Swallow Questionnaire; VCA, voluntary cough airflow; VFSS, videofluoroscopic swallow study; V-VST, volume-viscosity swallow test.
(doi: 10.1177/2040622318821622)
In the ESPEN guide (doi: 10.1016/j.clnu.2015.09.004), it ends up summarizing that since there are no specific tools for diagnosing patients with dementia that are used interchangeably with those of adults, this is equally valid.“As there are no differences in the screening process between
persons with and without dementia, we also refer to the corresponding recommendations in the ESPEN Guideline on Clinical
Nutrition and Hydration in Geriatrics [52].”
Anyway I create a new table for supplementary matterial
Table 6.Listing the most commonly used validated scales that can be used in dysphagia assessment for dementia patients;
a. The Oropharyngeal Dysphagia Screening Test for Patients and Professionals (ODS-PP) (Quirós S et al. 2020)b. The Eating Assessment Tool-10 (EAT-10) (Belasfky PC et al. 2008)c. The Swallowing Disturbance Questionnaire (SWAL-QOL) (McHorney CA et al 2000)d. The Caregiver Questionnaire – RaDID-QC (Duarte de Oliveira G et al. 2024)
3 y 4.
Taking into account the concepts of interdisciplinarity and algorithms, I add an explained sentence in the text, as you suggest, in the Introduction (Page 3, paragraph 2, line 11);
“Although the confirmatory diagnosis must be made as complete as possible, with techniques, tests, anthropometric examinations and blood tests, within an interdisciplinary team and also with the help of algorithms (25)”

Round 3
Reviewer 1 Report
Comments and Suggestions for Authors
Dear Authors,
thanks for your answers.
In the tables, decimal values use a period, while in the text they use a comma. It would be advisable to use the same symbol consistently throughout.
Author Response
For research article
Anchor-based and distributional responsiveness of the Spanish version of the Edinburgh Feeding Evaluation in Dementia Scale in older people with dementia: a longitudinal study
Response to Reviewer 1 Comments 3
|
||
1. Summary |
|
|
Thank you very much for taking the time to review this manuscript. Please find the detailed responses below and the corresponding revisions/corrections highlighted/in track changes in the re-submitted files.
|
2. Questions for General Evaluation |
Reviewer’s Evaluation |
Response and Revisions |
Does the introduction provide sufficient background and include all relevant references? |
Yes/Can be improved/Must be improved/Not applicable |
[Please give your response if necessary. Or you can also give your corresponding response in the point-by-point response letter. The same as below] |
Are all the cited references relevant to the research? |
Yes/Can be improved/Must be improved/Not applicable |
|
Is the research design appropriate? |
Yes/Can be improved/Must be improved/Not applicable |
|
Are the methods adequately described? |
Yes/Can be improved/Must be improved/Not applicable |
|
Are the results clearly presented? |
Yes/Can be improved/Must be improved/Not applicable |
|
Are the conclusions supported by the results? |
Yes/Can be improved/Must be improved/Not applicable
|
|
3. Point-by-point response to Comments and Suggestions for Authors
|
||
Comments 3: [Paste the full reviewer comment here.]
|
- Ensure all references are relevant to the content of the manuscript. done
- Highlight any revisions to the manuscript, so editors and reviewers can
see any changes made. done - Provide a cover letter to respond to the reviewers’ comments and
explain, point by point, the details of the manuscript revisions. done - If the reviewer(s) recommended references, critically analyze them to
ensure that their inclusion would enhance your manuscript. If you believe
these references are unnecessary, you should not include them. done - If you found it impossible to address certain comments in the review
reports, include an explanation in your appeal. done
If your manuscript requires improvement to the language and/or figures, you
may consider MDPI Author Services: https://www.mdpi.com/authors/english.
Does not apply
Please note the status of this invitation “Publish Author Biography on the
webpage of the paper” -
https://susy.mdpi.com/user/manuscript/author_biography/174a3b812a4e4946a59779181ea829dd.
If you wish to publish your biography, please complete it before your
manuscript is accepted.
Done
Comments and Suggestions for Authors
Dear Authors, thanks for your answers.
In the tables, decimal values use a period, while in the text they use a comma. It would be advisable to use the same symbol consistently throughout.
Response
Thank you very much.
Reviewed everything; text, tables and figures and in all numerical figures it has been expressed with a period. (see figure 3).
